# Impact of Sexual Abuse on Post-Traumatic Stress Disorder in Children and Adolescents: A Systematic Review

**Ana Carolina Alves** [1,2,†]**, Maria Leitão** [1,2,†]**, Ana Isabel Sani** [1,2,3] **and Diana Moreira** [2,4,5,6,7,*]

1    Faculty of Human and Social Sciences, Fernando Pessoa University, Praça 9 de Abril, 349,
     4249-004 Porto, Portugal; 38111@ufp.edu.pt (A.C.A.); 38098@ufp.edu.pt (M.L.); anasani@ufp.edu.pt (A.I.S.)
2    Observatory Permanent Violence and Crime (OPVC), FP-I3ID, Fernando Pessoa University,
     4249-004 Porto, Portugal
3    Research Center on Child Studies (CIEC), University of Minho, 4710-057 Braga, Portugal
4    Centro Regional de Braga, Universidade Católica Portuguesa, 4710-362 Braga, Portugal
5    Laboratory of Neuropsychophysiology, Faculty of Psychology and Educational Sciences, University of Porto,
     4200-135 Porto, Portugal
6    Centro de Solidariedade de Braga/Projeto Homem, 4700-024 Braga, Portugal
7    Institute of Psychology and Neuropsychology of Porto—IPNP Health, 4100-341 Porto, Portugal
*    Correspondence: dimoreira@ucp.pt
†    These authors contributed equally to this work.

**Abstract:** Child sexual abuse (CSA), the most common type of maltreatment, is any action of a sexual nature by one or more adults towards a minor without the minor's consent. This abuse represents one of the most damaging forms of trauma, has a severe impact on mental health and psychopathology, and can lead to several disorders, including post-traumatic stress disorder (PTSD). PTSD is characterized as a disorder that encompasses physical symptoms resulting from traumatic experiences that are experienced or witnessed by the victim. This systematic review aims to understand the impact of sexual abuse on post-traumatic stress disorder in children and adolescents. Studies focusing on the relationship between these two variables were obtained through multiple databases. Of the 940 documents collected, 24 were retained for further analysis and the objectives, methodologies, results, and main conclusions were registered. One of the main conclusions was that the earlier the abuse starts and the more severe and long-lasting it is, the symptomatology of PTSD will be aggravated and remain in the long term.

**Keywords:** child sexual abuse; post-traumatic stress disorder; minors; impact





## 1. Introduction

Sexual violence is characterized by a sexual game or act, which may occur in heterosexual or homosexual relationships, between one or more adults and a minor, with the purpose of using the minor to sexually stimulate themselves or attempting to sexually stimulate the minor. This type of violence comprises two specificities: sexual exploitation and sexual abuse, the last one being one of the core concepts of this systematic review (Florentino 2015).

Any action of sexual nature of one or more adults towards a minor is defined as sexual abuse, and this abuse can occur within the family, when the aggressor has affective ties and kinship with the minor, or outside the family, when the aggressor has no kinship with the child. The sexual abuse of children and adolescents represents one of the most frequent types of maltreatment (Florentino 2015).

Child sexual abuse (CSA) is a major international social problem. Worldwide statistics on this issue point to the worrying nature of the phenomenon. Through a meta-analysis of 217 publications published between 1980 and 2008, from different countries, it is estimated that the prevalence of sexual abuse in children under 18 years old is 18% in females and 7.6% in males (Cruz et al. 2021; Hébert et al. 2014). According to Cruz et al. (2021), in a

report compiling data from 2016 to 2017 from several countries in the United Kingdom, there were 54,846 reports of sexual violence against minors, with an expected increase in cases over the years. These same authors point to a recently implemented service in New Zealand, aimed at reporting and receiving victims of sexual abuse, in which 1200 contacts were recorded in just one month. It was found that most of the victims who contacted this service were between 13 and 19 years old.

Since sexual abuse represents one of the most harmful forms of trauma, having a severe impact on mental health and psychopathology, most published studies focus on the identification of long-term symptoms resulting from sexual abuse, using samples made up of adults to allow for retrospective studies (Hébert et al. 2014; Jardim et al. 2021).

Florentino (2015) describes that long-term symptoms may manifest themselves through the occurrence and incidence of psychiatric disorders such as affective dissociation, suicidal ideation, and acute phobias. Victims may present "more intense levels of fear, anxiety, depression, anger, guilt, isolation and hostility" (Florentino 2015, p. 141) and a chronic sense of danger and confusion, demonstrating distorted cognition and difficulty perceiving reality. It is possible that they may show illogical thinking, a decreased understanding of more complex roles, and difficulty in solving interpersonal problems. The author also mentions several manifestations that sexual abuse can take in the short term, such as fear of the aggressor or people of the same sex, symptomatic complaints, psychotic symptoms, social isolation and feelings of rejection, confusion, humiliation, shame, and fear. Anxiety-related phobias, obsessive–compulsive disorder, depression and/or sleep, learning, and eating disorders may also occur.

Reviews of the academic literature have concluded that sexual abuse in childhood represents a non-specific risk factor for a spectrum of psychological disorders, physical problems, and risky sexual behavior. Thus, the sexual abuse of minors leaves marks in different areas of the human condition—e.g., physical, sexual, psychological, and social marks—which may compromise the victim's life at different levels. The spectrum of psychological disorders mentioned above includes anxiety disorders, depression, suicidal ideation, and post-traumatic stress disorder (PTSD) (Hébert et al. 2014; Florentino 2015).

PTSD is a disorder that encompasses physical symptoms as a result of traumatic experiences that are experienced or witnessed by the victim, who repeatedly relives that past experience as it happened in the past (Cruz et al. 2021). According to the Diagnostic and Statistical Manual of Mental Disorders (DSM-5; APA 2013), PTSD is an anxiety disorder that follows exposure to a traumatic event. The victim's subjective assessment of this event involves dread and fear responses, and is interconnected with symptoms of reliving, avoidance, increased physiological excitability, and time and functional impairment (APA 2013).

People with PTSD present six common elements, these being (1) re-experiencing intrusive and persistent memories related to the trauma, (2) frequent exposure to various situations that lead the individual to remember the trauma, (3) the numbing of emotional reactions and the impediment of exposure to specific situations that are usually linked to the emotion of the trauma he/she experienced, (4) the decreased ability to use verbal language and the consequent replacement of this with gestures, (5) disorders related to lack of attention, and (6) changes in personal identity (Florentino 2015).

PTSD is the most common psychopathological condition associated with exposure to traumatic events in children and adolescents. In cases of child sexual abuse, the prevalence of PTSD varies between 20 and 70%, and girls tend to develop more symptoms than boys (Borges et al. 2010; Habigzang et al. 2010). Bernard-Bonnin et al. (2008) report that the frequency of PTSD in sexually abused children is high, since approximately 40% of abused children develop PTSD symptoms. Therefore, sexual abuse causes an increased risk of victims developing PTSD (Aded et al. 2006).

Sexual abuse in children and adolescents causes trauma which, in turn, has a major negative impact on the behavioral, social, and emotional development of those who are victims of this type of violence, affecting the cognitive, behavioral, and social areas. Taking

this into account and due to the stability of PTSD symptoms over time, to provide an adequate treatment, monitoring by several professionals is necessary (Araújo and Martins 2021; Bernard-Bonnin et al. 2008).

This systematic review aims to understand the impact of sexual abuse on post-traumatic stress disorder in children and adolescents. In order to achieve this main objective, it is important to elaborate specific objectives. This review explores (1) the proximity between victim and perpetrator, disclosure, and PTSD symptomatology, (2) how sexual abuse is perpetrated, (3) the age of first abuse, severity, and duration of CSA, (4) PTSD symptomatology in women who were victims of childhood sexual abuse, (5) brain changes after CSA, and (6) coping strategies. To this end, we focus on empirical studies that relate the variables under study, namely child sexual abuse and post-traumatic stress disorder.

According to Bernard-Bonnin et al. (2008), vulnerability to victimization and its consequences depend on the interaction between three mutually influencing variables. The first variable to be considered concerns personal characteristics, including age, stage of development, personality, coping strategies, and the relationship between the victim and the offender. Next, it is crucial to take into account the characteristics of the event, in this case sexual abuse, such as frequency, severity, duration, and whether violence was involved. Finally, attention must be paid to the variable of contextual factors, which includes the support system of the victim, community attitudes and values, and legal and care resources. The authors posit that a multidimensional analysis of protective factors may elucidate why certain children do not exhibit significant challenges. These factors are divided into three main groups: personal, family, and extra-family factors. The former relates to the coping strategies that the child puts in place and that can influence the consequences and outcomes of the abuse. The protective factors associated with family show that the adjustment of the child who has been a victim of sexual abuse is enhanced if there is family cohesion and positive support from the maternal figure.

## 2. Method

### 2.1. Article Search and Inclusion Criteria

The studies were collected through three databases, EBSCOhost, PubMed, and Web of Science, selecting only articles published in scientific journals and peer-reviewed journals. To define the search expression, an analysis of the keywords commonly used in articles addressing sexual abuse and PTSD was performed to cluster and gather the largest number of terms for the construction of the search expression.

### 2.2. Inclusion and Exclusion Criteria

To carry out this systematic review, inclusion and exclusion criteria were defined with the purpose of selecting relevant studies from the studies found.

Studies meeting the following inclusion criteria were included: (1) studies where the target population was children and adolescents, under 18 years of age, or an adult population but the sexual abuse was experienced under 18 years of age; (2) empirical studies; (3) studies addressing the relationship between CSA and PTSD.

Regarding exclusion criteria, we excluded all studies that were merely theoretical, other systematic reviews, studies that were not in Portuguese, English or Spanish, studies that referred to adults and seniors as target populations, and studies that did not address sexual abuse and PTSD.

The two researchers analyzed the survey results independently and discrepancies were resolved through a senior researcher to reduce the likelihood of losing any studies and to minimize errors that may have arisen in the process of ranking the studies.

## 3. Research Strategy

Before starting the search in the three databases used, we prepared the search expression, which subsequently underwent the necessary changes according to each of the databases. These changes resulted in two search expressions, one of them being used for

the search in two of the databases (EBSCO and PubMed), namely [AB (sexual abuse or sexual trauma or sexual violence or sexual assault or rape) AND AB "post traumatic stress disorder" AND AB (children or adolescents or youth or child or teenager)], and another specific to the Web of Science [AB = (sexual abuse or sexual trauma or sexual violence or sexual assault or rape) AND AB = "post traumatic stress disorder" AND AB =(children or adolescents or youth or child or teenager)].

The research conducted was only limited in terms of language, and no geographical or temporal boundaries were imposed.

The search in the three databases resulted in 940 studies, published between 1988 and 2024 (Figure 1). A total of 546 duplicates were excluded, leaving 394 articles for abstract analysis. After a careful analysis of these abstracts, 286 studies were excluded for the following reasons: being theoretical studies or written in a language other than Portuguese, English, or Spanish, using the wrong population, or presenting results which were not relevant to this systematic review. This exclusion resulted in 108 articles, of which 84 were excluded through full-text analysis; thus, the remaining 24 studies were included. The objectives, methodologies, results, and main conclusions were extracted from each of these 24 articles.

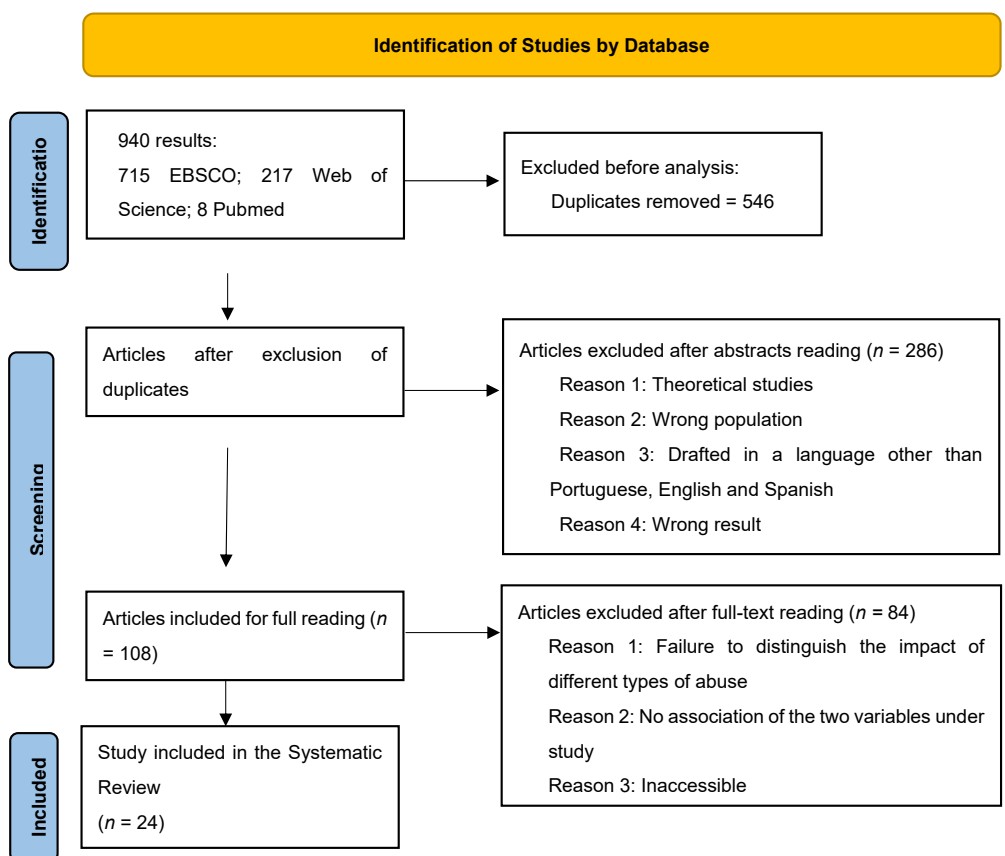

**Figure 1.** PRISMA 2020 flow diagram for new systematic reviews including searches of databases and registers only.

The Quantitative Research Assessment Tool (QRAT; Child Care & Early Education Research Connections n.d.) was used to assess the methodological quality of the studies included in this review. The QRAT comprises 12 items pertaining to the methodological features of studies. Items can be rated −1, 0, 1, or NA (not applicable), except for the 12th item, where NA is not an option. According to QRAT specifications, studies with lower scores should be regarded with more caution, compared to studies with higher scores, which are methodologically more robust. Most studies included in this review (61.5%) had a score of six or above (see Table 1).

**Table 1.** Article quality assessment.

| Study Id | Item 1 | Item 2 | Item 3 | Item 4 | Item 5 | Item 6 | Item 7 | Item 8 | Item 9 | Item 10 | Item 11 | Item 12 | Sum |
|---|---|---|---|---|---|---|---|---|---|---|---|---|---|
| Adams et al. (2018) | 0 | 0 | 0 | 1 | 1 | 1 | 1 | 1 | 1 | 1 | 1 | 1 | 9 |
| Aksu et al. (2018) | 0 | 0 | −1 | 1 | 1 | 1 | 1 | 0 | 1 | 1 | 1 | 1 | 7 |
| Bernard-Bonnin et al. (2008) | 0 | 0 | −1 | −1 | 1 | 1 | 1 | −1 | 1 | 1 | 1 | 1 | 4 |
| Bicanic et al. (2013) | 0 | 0 | −1 | 1 | 1 | 1 | 1 | 1 | 1 | 1 | 1 | 1 | 8 |
| Botsford et al. (2018) | 0 | 0 | −1 | 1 | 1 | 1 | 1 | 0 | 1 | 1 | 0 | 1 | 6 |
| Briggs and Joyce (1997) | 0 | 0 | −1 | 1 | 1 | 1 | 1 | −1 | 1 | 0 | −1 | 1 | 3 |
| Cantón-Cortés and Cantón (2010) | 0 | 0 | 1 | 1 | 1 | 1 | 1 | 1 | 1 | 1 | 1 | 1 | 10 |
| Cantón-Cortés et al. (2011) | −1 | 0 | −1 | 0 | 1 | 1 | 1 | 1 | 1 | 1 | 1 | 1 | 6 |
| Chiasson et al. (2021) | 0 | 0 | −1 | 0 | 1 | 1 | 1 | 0 | 1 | 1 | 1 | 1 | 6 |
| Daigneault et al. (2016) | 0 | 0 | −1 | 1 | 1 | 1 | 1 | 1 | 1 | 1 | 0 | 1 | 7 |
| Harding et al. (2012) | 0 | 0 | −1 | 1 | 1 | 1 | 1 | 1 | 1 | 1 | 1 | 1 | 8 |
| Lam (2015) | 0 | 0 | 0 | 1 | 1 | 1 | 1 | −1 | 1 | 1 | 1 | 1 | 7 |
| Lawyer et al. (2006) | 1 | 1 | 1 | 1 | 1 | 1 | 1 | −1 | 1 | 1 | 0 | −1 | 7 |
| Mathews et al. (2013) | 0 | 0 | −1 | 0 | 1 | 1 | −1 | −1 | 1 | 1 | 0 | 1 | 2 |
| Mutluer et al. (2017) | 0 | 0 | −1 | 1 | 1 | 1 | 1 | 0 | 1 | 1 | 1 | 1 | 7 |
| Rellini and Meston (2006) | 0 | 1 | −1 | 1 | 1 | 1 | 1 | −1 | 1 | 1 | 1 | 1 | 7 |
| Rinne-Albers et al. (2015) | 0 | 0 | −1 | 1 | 1 | 1 | 1 | 1 | 1 | 1 | 1 | 1 | 8 |
| Rinne-Albers et al. (2017) | 0 | 0 | −1 | 1 | 1 | 1 | 1 | 1 | 1 | 1 | 1 | 1 | 8 |
| Schneider et al. (2007) | 1 | 0 | −1 | 1 | 1 | 1 | −1 | −1 | 1 | 1 | 0 | 1 | 4 |
| Thompson et al. (2000) | 1 | 0 | −1 | 1 | 1 | 1 | −1 | −1 | 1 | 1 | −1 | −1 | 2 |
| Timmons-Mitchell et al. (1998) | 0 | 0 | −1 | 1 | 1 | 1 | 0 | NA | 1 | 1 | 1 | −1 | 4 |
| Vagni et al. (2021) | 0 | 0 | −1 | 1 | 1 | 1 | 0 | 0 | 1 | 1 | 0 | 1 | 5 |
| van den Bulk et al. (2016) | 0 | 0 | −1 | 1 | 1 | 1 | 1 | 1 | 1 | 1 | 1 | 1 | 8 |
| Yüce et al. (2015) | 0 | 0 | −1 | 1 | 1 | 1 | 0 | 0 | 1 | 1 | 1 | 0 | 5 |

## 4. Results

*4.1. Proximity between Victim and Aggressor, Disclosure, and PTSD Symptomatology*

In one of the included studies, the results showed that 11.5% of the victims of sexual abuse were victimized by a stranger (e.g., someone with whom the victim had never had any contact), 22.2% by a member of the family nucleus, 54.5% by someone known, and 11.9% by a stranger whom the victim did not know but recognized (e.g., someone the victim may have come across in an establishment, or on the street, but without close contact) (Lawyer et al. 2006). In turn, Mutluer et al. (2017) reach a more detailed conclusion, revealing that the abuser, in 62% of the victimizations, was the stepfather, stepbrother, or a relative. Lawyer et al. (2006) also stated that if the perpetrator was an acquaintance, or a stranger that was recognized by the victim or a family member, the adolescent remained at a higher risk of PTSD in the past 6 months, compared to adolescents sexually assaulted by a stranger.

Chiasson et al. (2021) obtained data that allowed the authors to realize that all the abuses reported by the sample were committed by men and, on average, the participants only disclosed the abuse at 23 years of age. Lam (2015) reports that negative feelings related to disclosure allow the prediction of victims' PTSD responses, rather than CSA severity. Bicanic et al. (2013) noted that all CSA victims in the study sample met the criteria for PTSD.

In the study by Mathews et al. (2013), three interviews were conducted over a 6-week period, and in the first interview more than two-thirds of the children presented PTSD symptoms. By the third interview, the results showed that the children who had total PTSD symptoms in the first interview continued to manifest PTSD and those who had partial symptoms showed a decrease.

Children and adolescents with PTSD as a result of CSA are presented in the study by Vagni et al. (2021) as more vulnerable to delayed and immediate suggestibility, i.e., post-traumatic stress disorder has significant power in all components of both types of suggestibility. In this study, children and adolescents exposed to traumatic events who presented with PTSD exhibited a higher performance in the main tasks studied and showed a tendency to incorporate suggestive information traits into their original memory (suggestibility) (Vagni et al. 2021).

*4.2. How Sexual Abuse Is Practiced*

Since sexual abuse corresponds to one of the variables under study in this systematic review, it is important to understand which form of abuse has the highest prevalence. Several studies show that sexual intercourse (vaginal, anal, oral penetration) is the most recurrent form of sexual abuse. Yüce et al. (2015) and Mutluer et al. (2017) state that sexual intercourse occurred in 48.9% and 68% of the cases of abuse studied, respectively. It can also be perceived, according to Yüce et al. (2015), that the abuser is usually male and, in 80.3% of cases, has a connection to the victim. Furthermore, the study by Yüce et al. (2015) states that penetration is the most common form of sexual abuse, perpetrated vaginally in the case of girls and anally in the case of boys.

*4.3. Age of First Abuse, Severity, and Duration of CSA*

Child abuse, whether sexual, physiological, or emotional, is associated with several problems, one of them being PTSD, which is more likely to occur when there is a history of abuse (Schneider et al. 2007). In fact, several studies support the idea that the age of first abuse and the duration and severity of CSA allow the prediction of greater PTSD symptomatology. Adams et al. (2018) argue that if abuse begins in a developmental period, be it early infancy, childhood, or adolescence, it is more associated with developing PTSD symptoms. Girls usually experience greater victimization by sexual abuse and more continued and severe abuse in these first three periods of life compared to boys and therefore experience more symptoms of depression, anxiety, and PTSD (Adams et al. 2018).

*4.4. PTSD Symptomatology in Women Who Were Victims of Childhood Sexual Abuse (CSA)*

Although the focus of this review is sexual abuse occurring before the age of 18 years, many women in adulthood present PTSD symptoms related to it (Briggs and Joyce 1997; Cantón-Cortés and Cantón 2010; Timmons-Mitchell et al. 1998; Thompson et al. 2000). Briggs and Joyce (1997) mention that when abuse is committed through sexual intercourse, it is associated with the severity of PTSD symptoms and, therefore, the association of CSA with PTSD symptoms remains significant, even when controlling for other variables such as the level of general symptomatology. A history of CSA and current PTSD increases the likelihood of women attempting suicide by six times compared to women with PTSD who have not been abused (Thompson et al. 2000).

Within this population of women who have suffered CSA, it is possible to perceive that there is a relationship between early maladaptive schemas and the existence of PTSD, since the adaptive schemas that these women have may prove to be ineffective in assimilating important information about the trauma, leading to possible overaccommodation in a negative sense (Harding et al. 2012). The negative beliefs that an individual has about themselves, about others, and about the world after a traumatic experience contribute to the development and maintenance of PTSD (Harding et al. 2012).

In the study by the aforementioned authors, it was possible to perceive in the group of women who had suffered from CSA that there are certain schemas more associated with the symptoms of PTSD. Schemas related to vulnerability to harm and distrust/abuse were important predictors for the diagnosis of PTSD. These schemas were defined as referring to beliefs that others may intentionally hurt, abuse, or humiliate us, and having an exaggerated belief that harm or catastrophe could occur at any moment (Young and Brown 1994 as cited in Harding et al. 2012). They also scored as predictors of PTSD, and schemas related to subjugation, social isolation, emotional inhibition, and abandonment (Harding et al. 2012).

Rellini and Meston (2006) conducted a study with a sample consisting only of women, dividing them into three groups: (1) women victims of CSA, (2) women victims of CSA diagnosed with PTSD, and (3) a control group. The second group, mentioned above, had lower cortisol levels compared to the third, and higher levels compared to the group of women victims of CSA without a diagnosis of PTSD. Bicanic et al. (2013) obtained results which are in line with the results of the abovementioned author, in that sexually abused individuals had lower cortisol levels compared to the control group.

Timmons-Mitchell et al. (1998) conducted an interesting study with mother–child dyads in which both mothers and children had been sexually abused. They found that mothers who had been victims showed more PTSD symptoms than non-abused mothers and showed fewer PTSD symptoms when they found out that their children had also been sexually abused. In turn, children of non-abused mothers also showed more PTSD symptoms after suffering CSA.

*4.5. Brain Alterations after CSA*

CSA causes several brain changes in its victims. Rinne-Albers et al. (2015), in their study, compared the impact of sexual abuse and neglect on the size of the corpus callosum in girls, with abuse having the greatest influence. In addition to this alteration, another observation in victims who have suffered child sexual abuse concerns the volume of gray matter in the brain. Rinne-Albers et al. (2017) empirically demonstrated that adolescents with PTSD after CSA had 14.8% less grey matter volume in the anterior cingulate cortex compared to the control group, consisting of individuals without PTSD. To contextualize this result, the aforementioned authors describe that the anterior cingulate cortex has as its main functions the regulation of blood pressure and heart rate.

Chronicity and the early onset of trauma may have implications for the relationship between the smaller right amygdala and PTSD symptoms (Mutluer et al. 2017). Regarding structural brain imaging, Mutluer et al. (2017) point out that PTSD has a bilateral impact on the brain. The connectivity between the hemispheres of people with PTSD proves to be

less efficient than in individuals without any kind of disorder, who show better symmetry, therefore suggesting better connectivity between the hemispheres.

At the amygdala level, it is possible to observe some alterations in groups of subjects showing PTSD resulting from CSA. This group was compared with two other groups made up, respectively, of subjects diagnosed with internalizing disorders and healthy individuals, that is, subjects without any kind of disorder. Through this comparison, an initial increase, similar in all groups, of amygdala activation—a brain region that has an important role in detecting the valence and intensity of emotional expression—and a relatively quick habituation to emotional faces was found (van den Bulk et al. 2016).

Adolescents with PTSD because of CSA demonstrated higher levels of self-reported fear of frightening faces compared to healthy subjects. However, when confronted with neutral and happy faces, there were no significant differences between the groups (van den Bulk et al. 2016).

When comparing the reaction times of patients with PTSD secondary to CSA with those diagnosed with internalizing disorders (depression and anxiety), the above authors obtained results showing that PTSD patients had longer reaction times. Since the scary faces, the happy faces, and the neutral faces appeared three times each, with 21 copies of each group of faces, it could be noticed that the reaction time for the first two scary faces was higher than that for the last two happy faces. Faces in stages 1 and 3 were neutral faces.

Across the three presentations, there was robust activation in the participants' right amygdala. If the first and last presentations are compared, it is possible to notice a significant activation in the bilateral amygdala, suggesting alterations in the level of amygdala activation over time. The left amygdala was only activated in individuals with PTSD due to CSA, suggesting differences in the habituation patterns of the left amygdala in the different study groups. The activation and habituation of the left amygdala were only significantly predicted when there was PTSD due to CSA (van den Bulk et al. 2016).

After the previous explanations, which were based on internal brain changes such as the amygdala, cortisol levels, and grey matter volume, the focus of the next result will be on working memory, this being a brain change with readily observable implications. The first analysis by Chiasson et al. (2021) revealed significant differences between participants who experienced sexual abuse and had PTSD and subjects who were abused but did not have PTSD in terms of brain activity during working memory. More specifically, the group that suffered CSA and had PTSD showed less activation in the cerebellum and in the left fusiform gyrus—which has important functions in the recognition of objects, faces, and facial expressions—compared to the other group.

*4.6. Coping Strategies*

Coping strategies and attributions may mediate the relationship between the characteristics of CSA and their influence on PTSD symptoms. On the other hand, the effects of attributions of guilt may be mediated by the strategies applied by the CSA victim (Cantón-Cortés et al. 2011). The study of these authors showed that coping strategies significantly moderated the relationship between CSA and PTSD symptomatology, with the avoidance strategy scoring significantly higher, i.e., this strategy was related to higher PTSD scores.

As Lazarus (1993) mentioned, coping strategies can be analyzed in terms of their function, i.e., being able to focus on the problem or emotion. In this way, coping strategies focused on the problem allow changing the stressful situation through actions centered on the environment or on oneself, and coping strategies focused on emotions, in turn, try to change the way one deals with the situation or the meaning of what has happened. Strategies focused on emotions include emotions of detachment, self-control, acceptance of responsibility, and avoidance, which may include behaviors of denial, self-destruction, distancing, or withdrawal (Brand and Alexander 2003).

Cantón-Cortés and Cantón (2010) state that the relationship between coping strategies and PTSD is stronger when the aggressor belongs to the victim's family, that is, when we

are dealing with intra-family abuse. On the other hand, there is a stronger relationship with PTSD when sexual abuse is continued.

The child's ability to avoid facing the problem and a low sense of hope are two variables that contribute to the prediction of PTSD symptoms. Bernard-Bonnin et al. (2008) found that children who demonstrated these two variables obtained higher PTSD scores in their study.

Daigneault et al. (2016) indicate that sexually abused adolescents report lower levels of mindfulness, compared to adolescents with no reports of sexual abuse, which explains the greater symptoms of PTSD.

When experiencing a traumatic event, there is confrontation with new information, which in most cases does not fit into the pre-existing cognitive schemas of the victim. In this way, the process of accommodation and assimilation occurs to try to integrate the information resulting from the event. Accommodation, in which the individual adjusts their pre-existing schemas to integrate new information, generally causes functional and adaptive beliefs; however, when overaccommodation occurs, there is an exaggeration, changing, for example, the belief "I am in control of what happens" to "I am never in control of what happens" (Resick and Schnicke 1992).

On the other hand, assimilation allows changing the information coming from the trauma to make it compatible with pre-existing schemas, leading to maladaptive beliefs because the feeling of control of what happens continues to be a schema of the victim. This leads to the information about the event being altered, producing the belief that if there is such control, then the victim is to blame for the event. Therefore, the maladaptive cognitions resulting from these two processes are called blocking points, which prevent the victim from integrating the traumatic experience and processing these events (Resick and Schnicke 1992). According to Harding et al. (2012), there is a higher risk of developing PTSD when the victim has a higher level of maladaptive schemas.

In their study, Botsford et al. (2018) found a significant relationship between a greater number of blocking points and longer CSA duration. The later the first episode of abuse occurs, according to Botsford et al. (2018), the more blocking points are presented, related to feelings of guilt and security. In this study, it was possible to perceive that female individuals who suffered CSA presented a greater number of blocking points in the categories of safety and guilt. In addition, people who were the target of penetration in sexual abuse had higher levels of blocking points in the fault category. It is also important to note that the average number of blocking points related to guilt was higher in individuals who were abused by a family member compared to individuals who were abused by someone belonging to their social context or a stranger. Finally, this study allows us to infer that the victims of CSA who have a greater number of blocking points in the confidence category have a greater severity of PTSD symptoms (Tables 2 and 3).

**Table 2.** Summary of the studies' characteristics.

| Author(s) (Year) | Objective(s) | Sample | Instruments | Results and Main Conclusions |
|---|---|---|---|---|
| Adams et al. (2018) | Estimate the commencement, severity, and duration of physical and sexual abuse experienced during childhood and adolescence, and comprehend the impact on mental health, including symptoms of depression, anxiety, and PTSD in young adulthood. | *N* = 1268 adolescents and young adults (mean age = 19.68 years) | Childhood Trauma Questionnaire; depression scale of the Diagnostic Interview Schedule for Children Predictive Scales; physiological anxiety scale of the Revised Children's Manifest Anxiety Scale (RCMAS); Child PTSD Symptom Scale | The findings showed that the beginning of physical abuse during middle childhood and the start of sexual abuse during middle childhood or adolescence were linked to various forms of psychopathology. However, the length and severity of physical or sexual abuse did not forecast psychopathology once the timing of onset was considered. Further analysis revealed that sexual abuse beginning in adolescence and its duration were predictive of anxiety and PTSD in females, respectively. Conversely, the severity of sexual abuse was associated with fewer PTSD symptoms in males. In general, abuse experienced after the age of 5 may have a more detrimental effect on mental health. |
| Aksu et al. (2018) | The present study aims to compare the levels of brain-derived neurotrophic factor (BDNF), proBDNF, and tissue plasminogen activator (tPA) in individuals who developed PTSD because of sexual abuse, and the possible correlations with the severity of the symptoms. | *N* = 62 (31 female adolescents aged 8–18 years (with PTSD due to sexual abuse) and 31 females without PTSD) | Clinician-Administered Post-Traumatic Stress Disorder Scale for Children and Adolescents (CAPS-CA) | Serum BDNF and proBDNF levels in the group with PTSD were significantly lower, but the tPA level was significantly higher compared to healthy control subjects. There were no correlations between CAPS-CA scores and BDNF, or proBDNF and tPA levels. Low BDNF levels, which are suggested to be related to the etiopathogenesis of PTSD, appear to be a result of reduced proBDNF production. On the other hand, increased tPA levels in such cases may be a compensatory mechanism that serves to increase the levels of BDNF. |

**Table 2.** *Cont.*

| Author(s) (Year) | Objective(s) | Sample | Instruments | Results and Main Conclusions |
|---|---|---|---|---|
| Bernard-Bonnin et al. (2008) | To analyze the predictive factors of post-traumatic stress disorder (PTSD) symptoms in school-aged girls. | A group (*n* = 67) of girls aged 7 to 12 years and a group (*n* = 67) of non-abused girls | Family Relationship Index; Indice de Détresse Psychologique Enquête Santé Québec (IDPESQ); Children's Impact of Traumatic Events Scale—Revised; Self-Report Coping Scale; Children's Attributional Style Questionnaire—Revised; Self-Perception Profile for Children; Relationship Inventory; Perceived Social Support Scale | Higher prevalence of PTSD in abused girls. In addition, the child's perception of parental support and the child's confidence in avoidant coping predict PTSD symptoms. As predictive factors of PTSD are related to both child variables and family context variables, intervention should target both. |
| Bicanic et al. (2013) | To assess HPA axis (hypothalamic–adrenal–pituitary axis) function in female adolescents with rape-related PSPT but no prior sexual trauma compared to non-victimized controls. | *N* = 89 (52 adolescent rape victims with PTSD and 37 healthy adolescent females) | Interview; self-report questionnaires; Anxiety Disorders Interview Schedule—Children's version (ADIS-C); DSM-IV-based semi-structured clinical interview | In comparison to age-matched controls, adolescent rape victims with PTSD exhibited notably lower levels of cortisol and DHEAS. There were no discernible differences between groups with regard to the suppressive effect of dexamethasone. Both the rape event and the diagnosis of PTSD, and not factors such as sleep duration, smoking, education, or oral contraceptives, were responsible for the difference in neuroendocrine rates between rape victims and controls. |
| Botsford et al. (2018) | To evaluate the post-traumatic maladaptive cognitions, the so-called "stuck-points", of forty-three adolescent survivors of interpersonal trauma. | *N* = 43 individuals with an average age of 17.3 years | Basic Documentation of Trauma Characteristics; Clinician-Administered PTSD Scale, Children and Adolescent Version (CAPS-CA); University of California Los Angeles PTSD Reaction Index (UCLA); Beck Depression Inventory (BDI-II) | Sexual abuse was associated with "blocking points" in the safety and guilt domains. Factors such as penetration, female gender, having suffered the trauma at an older age and having a closer relationship with the perpetrator were associated with the guilt domain. Finally, a higher number of stuck-points in the trust category was related to greater severity of PTSD symptoms. Therapists should pay attention to these different themes, with the aim of providing the best possible treatment for each individual patient. |

**Table 2.** *Cont.*

| Author(s) (Year) | Objective(s) | Sample | Instruments | Results and Main Conclusions |
|---|---|---|---|---|
| Briggs and Joyce (1997) | To determine which childhood abuse experiences are associated with post-traumatic stress disorder (PTSD) symptomatology for female survivors of child sexual abuse (CSA). | *N* = 73 women | Self-report questionnaires; Hopkins Symptom Check List (SCL-90); Social Adjustment Scale (SAS) | The sample consisted of 73 women, aged 31.5 (*SD* = 7.8) years, who had experienced sexual abuse in childhood. For most women, the abuse occurred between the ages of 11 and 15. The first finding was that many women who experienced childhood sexual abuse presented with the current symptoms of PTSD. Second, the severity of PTSD symptoms correlated with the extent of general psychopathology. Third, the severity of PTSD symptoms was associated with the extent of the post-traumatic stress and whether the abuse involved actual sexual intercourse. |
| Cantón-Cortés and Cantón (2010) | To evaluate the positive or negative impacts of coping strategies employed by victims of childhood sexual abuse on the symptomatology of PTSD. | *N* = 276 (138 victims of child sexual abuse and 138 participants for the control group) | Questionnaire on child sexual abuse; How I Deal With Things Scale; Severity of Symptoms of PTSD Scale | Victims of sexual abuse exhibited higher PTSD scores and lower coping strategies. Nonetheless, discrepancies in prevention strategies across groups were inconsistent and did not consistently align with anticipated patterns. Only the implementation of prevention strategies correlated with PTSD, with participants utilizing them demonstrating heightened scores. The effects of prevention strategies were more pronounced in cases of ongoing abuse compared to isolated incidents, within familial abuse compared to non-familial, and among victims of sexual abuse versus non-victims. |

**Table 2.** *Cont.*

| Author(s) (Year) | Objective(s) | Sample | Instruments | Results and Main Conclusions |
|---|---|---|---|---|
| Cantón-Cortés et al. (2011) | To analyze, in a sample of CSA victims, the relationships between the severity of abuse, the attribution of blame, coping strategies, the existence of other maltreatment, and PTSD. | $N$ = 163 female university students | Child Sexual Abuse Questionnaire; The Attributions of Responsibility and Blame Scale; How I Deal With Things Scale; Post-traumatic Stress Disorder Symptom Severity Scale | The highest correlation was between avoidant symptoms and hyperactivation, followed by correlations with avoidant symptoms and re-experiencing. Although studies on CSA suggest that abuse characteristics may have an influence on PTSD symptomatology, this relationship may be entirely mediated by attributions and coping strategies. On the other hand, as the literature on sexual assault victims in particular suggests, the effects of blame attributions may be entirely mediated by the strategies employed by the victim. |
| Chiasson et al. (2021) | To contribute with a preliminary insight into the neural basis of the impact of CSA during two working memory tasks. | $N$ = 29 (16 victims of ABI and 13—control group—men) | The Childhood Trauma Questionnaire (CTQ); The Sexual Victimization Survey; SCID; Wechsler Adult Intelligence Scale (WAIS-III) | Preliminary analysis between participants with (CSA + PTSD) and (CSA + no PTSD) revealed a significant difference in brain activity during working memory. Specifically, the (CSA + PTSD) group showed less activation in the cerebellum and the left fusiform gyrus compared to (CSA + no PTSD). Two significant clusters were identified where participants with histories (CSA + no PTSD) had significantly less activation than control participants for the 2 working memory conditions. |
| Daigneault et al. (2016) | To examine whether mindfulness serves as a mediator and moderator in the association between self-reported exposure to childhood sexual abuse (CSA) and post-traumatic symptoms during adolescence. | $N$ = 245 adolescents (48% female) | Child Self-Acceptance, Mindfulness Measure (CAMM); Trauma Symptoms Checklist for Children (TSCC) | Mindfulness serves as a mediator of post-traumatic symptoms, as CSA correlates with reduced levels of mindfulness, which in turn correlates with increased post-traumatic symptomatology. However, mindfulness only serves as a moderating factor for CSA in relation to anger and anxiety, but not in the anticipated manner. |

**Table 2.** *Cont.*

| Author(s) (Year) | Objective(s) | Sample | Instruments | Results and Main Conclusions |
|---|---|---|---|---|
| Harding et al. (2012) | To examine the relationship between early maladaptive schemas and PTSD symptoms among a sample of women with histories of childhood sexual abuse. | *N* = 127 women with a history of CSA and 50 with no history | Childhood Trauma Questionnaire (CTQ); Young Schema Questionnaire-Short Form (YSQ-S); Purdue PTSD Scale—Revised (PPTSD-R); Beck Depression Inventory-II (BDI-II); Trauma Symptom Inventory (TSI) | A discriminant analysis indicated that the distrust/abuse, vulnerability to harm, and emotional deprivation schemas contributed most to distinguishing women based on presumed PTSD diagnostic status. The results highlight the importance of cognitive factors in the development and/or maintenance of PTSD symptoms and suggest possible treatment targets for cognitive therapy with CSA survivors. |
| Lam (2015) | To explore the impact of various factors on child sexual abuse, in particular the role of disclosure experience after controlling for other factors. | *N* = 800 students aged between 13 and 16 years(408 were female) | Distress Disclosure Index (DDI); Children's Attribution and Perception Scale (CAPS); Inventory of Parent and Peer Attachment—Revised (IPPA-R); Culture-FreeSelf-Esteem Inventories (CFSEI-2); Children's Impact of Traumatic Events Scale—Revised (CITES-R); Child Behavior Checklist—Youth Self Report (CBCL-YSR); Children's Revised Impact of Event Scale (CRIES) | Participants with a history of child sexual abuse reported significantly poorer psychological well-being than participants without ASI. The majority of participants experienced sexual abuse perpetrated by individuals outside of their family, such as friends or strangers. This study revealed that feelings of negativity towards disclosing the abuse, rather than the severity of the abusive experience, predicted responses to disclosure-related PTSD. |
| Lawyer et al. (2006) | To examine mental health outcomes as a function of the victim–aggressor relationship among interpersonally assaulted adolescents. | *N* = 4023 adolescents | Structured interview | More than half of the sample was assaulted by a known individual. Adolescents sexually assaulted by someone they knew, someone they did not know but recognized, or a family member remained at increased risk for PTSD in the past 6 months compared to adolescents assaulted by strangers. |

**Table 2.** *Cont.*

| Author(s) (Year) | Objective(s) | Sample | Instruments | Results and Main Conclusions |
|---|---|---|---|---|
| Mathews et al. (2013) | To assess the psychological adjustment of children after sexual assault. | *N* = 50 female children aged 8 to 17 years old | Semi-structured interviews with caregivers, structured interviews with children, Child Depression Inventory (CDI), Child Manifest Anxiety Scale, Child PTSD Checklist | At the first interview, just over two-thirds of the children had symptoms indicative of PSPT and only one was asymptomatic in the period immediately after disclosure. At the second interview, PTSD symptoms decreased, but 90% of children still met the combined criteria of full and partial symptoms. At the third interview, the proportion of children with full symptomatic PTSD remained unchanged. |
| Mutluer et al. (2017) | To investigate the neurobiological response to stress and its clinical correlates among adolescents with PTSD, adding the investigation of the volume of selected brain regions compared to the control group, absent from PTSD. | *N* = 44 (23 adolescent girls with PTSD and 21 adolescents—control group) | Clinician-Administered PTSD Scale for Children and Adolescents (CAPS-CA); Adolescent Dissociative Experiences Scale (ADES); Schedule for Affective Disorders and Schizophrenia for School Age Children—Present and Lifetime Version (KSADS-PL); Beck Depression Inventory (BDI); Childhood Trauma Questionnaire (CTQ); Childhood Abuse and Neglect Questionnaire (CANQ); Neuropsychological Tests (Tower of London, Clock Drawing, Judgement of Line Orientation, Stroop TBAG, and Serial Digit Learning) | For 14 (68%) of the participants the sexual abuse included coitus, for 9 (49.9%) of them the abuse occurred regularly, and for 14 (62%) the predator was the stepfather, stepbrother or a relative or acquaintance imposed by the parents. These findings highlighted a bilateral effect of PTSD on the brain, with the right hemisphere playing a predominant role in secondary modes of reaction. However, the central symptom of dissociation appeared to be related to the left brain, in particular the left prefrontal cortex. The amygdala was the structure most consistently implicated. The relationship between the smaller right amygdala and PTSD symptoms can be attributed to the chronicity and early onset of the trauma. |

**Table 2.** *Cont.*

| Author(s) (Year) | Objective(s) | Sample | Instruments | Results and Main Conclusions |
|---|---|---|---|---|
| Rellini and Meston (2006) | To understand the impact of SNS activation through intense exercise on sexual arousal in women with CSA and PTSD. | *N* = 39 women | Sociodemographic questionnaire, two videos, Clinician-Administered Post-traumatic Stress Disorder Scale (CAPS), Female Sexual Function Index (FSFI) and Life Stressor Checklist Revised (LSC-R) | The FSFI showed significantly lower levels of arousal from functioning and women with CSA+PTSD compared to controls. However, there was no statistically significant difference between the groups in terms of how women reported reacting to erotic material. Physiological sexual arousal, continued sexual arousal, and lower cortisol levels were found in women with CSA+PTSD compared to the control group. |
| Rinne-Albers et al. (2015) | To ascertain whether there are discrepancies in white matter integrity in the brains of adolescents with PTSD stemming from CSA compared to healthy adolescents, and to explore potential correlations between white matter integrity and symptom severity within the patient group. | *N* = 40 adolescents | Trauma Symptom Checklist for Children (TSCC); Anxiety Disorders Interview Schedule Child and Parent Versions (ADIS-C/P); Puberty Development Scale (PDS) | The group with PTSD exhibited notably elevated scores across all TSCC scales. Additionally, the PTSD group demonstrated reduced FA (fractional anisotropy) values in the genu, midbody, and splenium of the CC (corpus callosum). The research indicated significant discrepancies in FA values between the groups, with a marked elevation in RD (radial diffusivity) and MD (mean diffusivity) observed in the PTSD group compared to the control group. |
| Rinne-Albers et al. (2017) | To investigate abnormalities in grey matter volume in a homogeneous group of adolescents with PTSD due to child sexual abuse and symptom severity. | *N* = 46 adolescents | Anxiety Disorders Interview Schedule Child and Parent Versions (ADIS-C/P); Trauma Symptom Checklist for Children (TSCC); Adolescent Dissociative Experiences Scale (A-DES) | On average, adolescents with CSA-related PTSD exhibited 14.8% lower grey matter volume in the anterior cingulate cortex compared to healthy non-traumatized controls. Exploratory whole-brain analysis revealed no differences in grey matter volume between patients and controls. |

**Table 2.** *Cont.*

| Author(s) (Year) | Objective(s) | Sample | Instruments | Results and Main Conclusions |
|---|---|---|---|---|
| Schneider et al. (2007) | To assess the independent risk of adult mental health problems, including likely diagnosis of PTSD, associated with each type of child abuse (i.e., sexual, physical, and emotional). | *N* = 3936 women | California Women's Health Survey (CWHS); Traumatic Stress Schedule (TSS); Centers for Disease Control and Prevention (CDC); PC-PTSD | Sexual, physiological, and emotional child abuse were each associated with a significant risk of frequent mental distress, frequently overwhelmed feelings, frequent anxiety, frequent sadness, and probable PTSD diagnosis. The risk of a probable PTSD diagnosis was especially high. |
| Thompson et al. (2000) | To examine whether the five forms of child maltreatment and PTSD have an individual or combined impact on the prediction of suicide attempts. | *N* = 335 African American women | National Women's Study (NWS); Childhood Trauma Questionnaire (CTQ) | Women in the attempt group were markedly more prone to manifest PTSD compared to those in the control group, and they were also significantly more inclined to report experiencing all five forms of child maltreatment. Notably, CSA without concurrent PTSD did not heighten a woman's risk of engaging in suicidal behavior. |
| Timmons-Mitchell et al. (1998) | To explore the relationship between children's allegations of sexual abuse, children's PTSD symptoms, mothers' histories of sexual abuse, and mothers' PTSD symptoms in the post-disclosure period. | *N* = 28 (14 mother-child pairs) | Child Behavior Checklist (CBCL); Structured Pediatric Psychosocial Interview (SPPI); SCL-90-R; Purdue Post-traumatic Stress Disorder Scale (PPTSD-R) | Differences were found between children of sexually abused and non-sexually abused mothers, but the differences were in an opposite direction to what was predicted. Children of non-abused mothers showed more PTSD symptoms than children of abused mothers. It may be that women abused in childhood who show symptoms of PTSD repress their own avoidance when their children report sexual abuse. Mothers who reported being victims of sexual abuse revealed significantly more PTSD symptoms than non-abused mothers. |

**Table 2.** *Cont.*

| Author(s) (Year) | Objective(s) | Sample | Instruments | Results and Main Conclusions |
|---|---|---|---|---|
| Vagni et al. (2021) | To examine whether and how memory tasks and levels of suggestibility are affected by PTSD in children called to testify in cases of suspected sexual abuse. | *N* = 114 children and adolescents aged between 8 and 16 years | Trauma Symptom Checklist for Children (TSCC); Gudjonsson Suggestibility Scale 2 (GSS2) | The findings indicate that children and adolescents with PTSD are more susceptible to immediate suggestibility. The most notable differences were observed after the initial interview, particularly following negative feedback and the repetition of leading questions. A comparison between participants with and without PTSD unveiled significant discrepancies in total suggestibility and yield. |
| van den Bulk et al. (2016) | To examine habituation patterns of amygdala activity to emotional faces (fearful, happy, and neutral) in adolescents with a depressive and/or anxiety disorder, adolescents with ASI-related PSPT, and healthy control group. | *N* = 71, 26 with a diagnosis of depressive and/or anxiety treatment, 26 controls, and 19 with CSA-related PTSD. | Anxiety Disorders Interview Schedule (ADIS); Child Depression Inventory (CDI); Revised Children's Anxiety and Depression Scale (RCADS); Trauma Symptom Checklist for Children (TSCC) | Adolescents with internalized disorders exhibit only partial similarity in symptoms compared to adolescents with CSA-related PTSD. Healthy adolescents demonstrated a habituation effect in the amygdala when exposed to emotional faces. The findings of these analyses revealed that activation and habituation of the amygdala, particularly on the left side, were primarily influenced by the presence of ASI-related PTSD. |
| Yüce et al. (2015) | To explore the psychiatric outcomes of sexual abuse and its related factors in children and adolescents referred to their child and adolescent psychiatry clinic from official medico-legal units. | *N* = 590 individuals with ages between 1 and 18 years old (507 girls and 83 boys) | Wechsler Intelligence Scale for Children—Revised Form (WISC-R); Schedule for Affective Disorders and Schizophrenia for School Age Children—Present and Lifetime Version—Turkish Version (K-SADS-PL-T) | Sexual intercourse (vaginal/anal penetration and/or being subject to prostitution) occurred in 48.9% of the cases. All perpetrators were male, and in 80.3% of the cases they were related to the victims [incest, *n* = 91 (15.1%)]. The most prevalent psychiatric disorders associated with sexual abuse were as follows: depressive disorder (45.9%), PTSD (31.7%), acute stress disorder (11.5%), anxiety disorder (1.1%), and conversion disorder (1.1%), in descending order. A total of 143 patients (24.2%) were not diagnosed with any psychiatric disorder. |

**Table 3.** Critical findings.

| Study ID | Critical Findings |
| --- | --- |
| Adams et al. (2018) | Increased likelihood of developing PTSD symptoms is associated with greater precocity, severity, and duration of abuse. |
| Aksu et al. (2018) | Adverse childhood experiences in which sexual abuse is included are potential triggers for impact on executive functions. |
| Bernard-Bonnin et al. (2008) | Sexual abuse causes an increased risk of victims developing PTSD. Children's ability to avoid facing the problem and low sense of hope contribute to higher PTSD scores. |
| Bicanic et al. (2013) | Sexually abused individuals had lower cortisol levels compared to the control group. |
| Botsford et al. (2018) | Significant relationship between higher number of blocking points and longer CSA duration. |
| Briggs and Joyce (1997) | Abuse committed through sexual intercourse is associated with the severity of PTSD symptoms. |
| Cantón-Cortés and Cantón (2010) | Victims of CSA, compared to those who have not suffered this form of victimization, show greater difficulties in adjusting in the long term. |
| Cantón-Cortés et al. (2011) | Coping strategies significantly moderate the relationship between CSA and PTSD symptomatology. |
| Chiasson et al. (2021) | There are significant differences between participants who have experienced sexual abuse and have PTSD and subjects who have been abused but do not have PTSD in brain activity during working memory. |
| Daigneault et al. (2016) | Sexually abused adolescents report lower levels of mindfulness compared to adolescents with no reports of sexual abuse, which explains greater symptoms of PTSD. |
| Harding et al. (2012) | The negative beliefs that the individual holds about themselves, others, and the world in the aftermath of a traumatic experience are a contributing factor to the development and maintenance of PTSD. |
| Lam (2015) | Negative feelings related to disclosure allow prediction of victims' PTSD responses, rather than CSA severity. |
| Lawyer et al. (2006) | If the perpetrator was an acquaintance, unknown but recognized by the victim or a family member, the adolescent remained at greater risk of experiencing PTSD in the past 6 months compared to adolescents sexually assaulted by a stranger. |
| Mathews et al. (2013) | The level of PTSD symptoms displayed by the victims after 4-5 months of the abuse was still significant, which may suggest a worrying long-term psychological adjustment. |
| Mutluer et al. (2017) | The connectivity between the hemispheres of people with PTSD proves to be less efficient than in individuals without any kind of disorder, who show better symmetry, therefore suggesting better connectivity between the hemispheres. |

**Table 3.** *Cont.*

| Study ID | Critical Findings |
| --- | --- |
| Rellini and Meston (2006) | Women victims of CSA with a diagnosis of PTSD had lower cortisol levels compared to the control group and higher levels compared to the group of women victims of CSA without a diagnosis of PTSD. |
| Rinne-Albers et al. (2015) | The study compared the impact of sexual abuse and neglect on the size of the corpus callosum in girls, with abuse showing the greatest influence. |
| Rinne-Albers et al. (2017) | Adolescents with PTSD after CSA showed 14.8% less grey matter volume in the anterior cingulate cortex compared to the control group consisting of individuals without PTSD. |
| Schneider et al. (2007) | Child abuse, whether sexual, physiological, or emotional, is associated with various problems, one of which is PTSD, which is more likely to occur when there is a history of abuse. |
| Thompson et al. (2000) | A history of CSA and current PTSD increases the likelihood of women committing a suicide attempt by six times compared to women with PTSD but who have not been abused. |
| Timmons-Mitchell et al. (1998) | Mothers who have suffered from sexual abuse show greater symptomatology regarding the abuse they have suffered, but less PTSD symptomatology regarding the abuse their children have suffered. |
| Vagni et al. (2021) | PTSD has significant power in all components of the two types of suggestibility (delayed and immediate). |
| van den Bulk et al. (2016) | Adolescents manifesting PTSD, arising from CSA, demonstrated higher levels of self-reported fear towards frightening faces compared to healthy individuals. |
| Yüce et al. (2015) | Penetration is the most common form of sexual abuse, vaginally in the case of girls and anally in the case of boys. |

## 5. Discussion

This systematic review aims to understand the impact of sexual abuse on post-traumatic stress disorder in children and adolescents through the achievement of specific objectives: (1) understanding the proximity between victims and perpetrators, disclosure, and PTSD symptomatology, (2) identifying how sexual abuse is perpetrated, (3) considering the age of first abuse, severity, and duration of CSA, (4) analyzing PTSD symptomatology in women who were victims of childhood sexual abuse, and understanding (5) brain changes after CSA and (6) coping strategies.

The proximity between victims and aggressors proved to be a significant risk factor for mental health symptoms, specifically for PTSD, among sexually assaulted adolescents. There are possible reasons that may explain the severity of PTSD symptoms related to the proximity between the aggressor and the victim. The first reason is that when the abuser is known, it may affect the beliefs of trust and intimacy that the victim had towards the abuser. If the abuser were a stranger, this shaking of beliefs might not occur so significantly. Next, it is possible that adolescents who are abused by acquaintances are at greater risk for PTSD due to other environmental factors that are not directly associated with aggression (e.g., living in a recurrent abusive environment). A third explanation is the fact that adolescents who suffer sexual abuse by an acquaintance, whether a family member or not, are less

likely to report the incident. This may cause them to have less social and psychological support, thus increasing the risk of symptoms (Lawyer et al. 2006).

As Lam (2015) previously mentioned, disclosure may be associated with negative feelings due to the fact that it is related to low social support from those close to the victim, and a possible blaming or self-blaming of the victim, enabling higher PTSD outcomes.

After Mathews et al. (2013) conducted the three interviews, it was noticed that the level of symptoms presented by the victims was still significant 4 to 5 months after the abuse, which may suggest a worrying long-term psychological adjustment.

Regarding the age of the first incident, severity, and duration of CSA, studies have concluded that the earlier the abuse occurs, and the more severe and long-lasting it is, the greater the likelihood of developing PTSD symptomatology (associated with possible depression and anxiety). Adams et al. (2018) point to a prediction of greater PTSD symptomatology in women related to the duration and severity of abuse. Conversely, men show a lower prediction of PTSD symptomatology in relation to the prolongation of abuse and increased severity of abuse, which may be explained by the greater stigmatization of sexual abuse suffered by males, making victims less willing to disclose psychological symptoms.

According to Adams et al. (2018), sexual abuse occurring before the age of 6 years does not predict PTSD symptoms. The authors of the present review defend this statement because they believe that the impact of abuse will be greater on adolescents as they are fully aware of what happened and are not deceived by the possible excuses and explanations given by the abuser.

For women who were abused in their childhood, subjugation, social isolation, emotional inhibition, and abandonment schemas collectively reflect a passive interpersonal style, in which women view themselves as having flaws, being inadequate, or different from others, and start to adopt a tendency to inhibit emotional expression (Harding et al. 2012).

One of the most interesting findings of the study by the aforementioned authors is related to the fact that, when comparing women victims of CSA who exhibited low levels of maladaptive schemas with women who had not been victims of this type of abuse, the female victims of CSA showed higher levels of abuse. However, regarding PTSD symptoms or maladaptive schemas, there were no differences between the two groups, which may suggest that there are protective factors that condition the influence of negative experiences which occur in childhood on the development of maladaptive cognitive schemas. Protective factors may include social support, positive early relationships, secure attachment to caregivers, or even individual differences in personality (Harding et al. 2012).

Timmons-Mitchell et al. (1998) conducted a study with mother–child dyads in which both had been sexually abused. Mothers who had been sexually abused were found to have higher symptomatology regarding the abuse they had suffered, but lower PTSD symptomatology regarding the abuse their children had suffered. These mothers scored high on the PTSD avoidance scale, which differentially affected their ability to register the PTSD symptoms shown by their children. In turn, these children showed a tendency to feel very attached to their mother, hence the lower symptomatology shown by them and their greater adjustment after the abuse.

As the authors note, adverse experiences that occur in childhood, for instance sexual abuse, are potential causes of impact at the level of executive function, as is found in the case of working memory, and at the level of emotional regulation, as is found in the case of the emotional identification that occurs in the amygdala (Aksu et al. 2018; Chiasson et al. 2021).

van den Bulk et al. (2016) focused their study on observing amygdala habituation patterns in response to emotional faces. Based on the assumption that the amygdala evaluates the positive or negative character of a stimulus and that its stimulation generates anxiety and fear, the authors showed that adolescents who suffered CSA presented a high level of fear in the face of fearful faces, just like adolescents with depression and anxiety. However, victimized adolescents had a slower response to emotional facial expressions.

At the neurological level, adolescents with PTSD due to CSA showed a greater activation of the amygdala at the beginning of the task, compared to the other study groups, but

by the end of the task, they showed similar levels of activation. This decrease in the level of activation may demonstrate a relatively rapid habituation of the amygdala, which means that the amygdala in individuals with PTSD due to CSA decreases, over time, its response to the stimulus that it considered to be irrelevant.

There are several interpretations for the different habituation effects among adolescents with CSA-related PTSD, for example, while the integration of information occurs intact in the cognitive control regions, greater stimulation may result in greater vigilance towards information that comes from the outside, which causes an increased response in the amygdala, which gets used to it over time (van den Bulk et al. 2016).

Working memory is an important cognitive faculty for daily basic functioning and work-related activities. In this way, working memory implies focus and sustained attention, which proves to be important for learning, solving problems, reasoning, and other faculties that are important for successful day-to-day functioning.

Individuals with PTSD resulting from CSA may experience intrusive thoughts resulting from internal or external triggers, causing a decrease in the accuracy of working memory in tasks that arise in the individual's day-to-day life, with attention deviating from the task (Chiasson et al. 2021). Negative emotional stimulation can be evidenced by a greater activation of the limbic system, and constitutes important evidence to help understand the potential long-term consequences of CSA in men, since there is an increased risk of suicidal behavior when it exists, a lower working memory capacity, and lower executive function scores (Chiasson et al. 2021).

Men who have suffered from CSA, with or without PSPT diagnoses, are affected by negative emotional stimuli during working memory. Clinicians can help these subjects learn to process the negative emotions arising from the abuse in a way that does not interfere with working memory processing, for example, using mindfulness training. Nowadays, men have a low incidence of disclosing abuse, due to society's prejudice and the masculinity imposed on men. The latter potentiates attributions of self-responsibility, influencing the way men deal with the experience of abuse. Empirical evidence of increased limbic activation during working memory influenced by negative emotion may help explain previously misunderstood behaviors by male victims. This knowledge may help reduce the stigma associated with this issue and provide additional information for clinicians to help these victims.

Cantón-Cortés and Cantón (2010) support the conclusions of other studies that show greater difficulties of long-term adjustment in CSA victims than in those who have not suffered this form of victimization. Therefore, these authors focused their study on trying to infer the positive or negative influence of PTSD symptomatology on victims' coping strategies. First, they found that victims with PTSD struggle with using coping strategies due to poor psychological adjustment. Those who suffer from CSA show lower levels of nervous coping and higher levels of avoidant coping. The latter type of coping and self-destructive coping lead to an increased risk of PTSD in young people with a history of CSA. Coping strategies proved to be the only strategies in which a clear difference was found between victimized and non-victimized individuals in the expected direction. In other words, this difference supports the hypothesis of victims' tendencies to have a greater use of maladaptive coping, explaining the relationship between coping and PTSD.

Avoidant coping is a strong predictor of PTSD and, according to Bernard-Bonnin et al. (2008), this finding may be explained by the fact that the behavioral and psychological consequences seem to share common characteristics with PTSD symptomatology (re-experiencing, avoidance, and hyperactivation). In situations of avoidant coping, another PTSD diagnostic criterion is present, namely avoidant behaviors and affective numbing.

Daigneault et al. (2016) showed that sexually abused subjects showed lower levels of mindfulness than those who had not been abused, and this explains why they show more PTSD symptoms. However, when they manage to develop higher levels of mindfulness, they also show higher levels of anxiety and anger, since higher levels of mindfulness

comprise a greater acceptance of the victimized individual's own feelings and thoughts. Thus, it is at lower levels of mindfulness that CSA has less effect on anxiety and anger.

With regard to blocking points, CSA is significantly related to guilt and safety blocking points. A possible explanation for the emergence of feelings of guilt after CSA may be the prejudice held by many people that the victim was partially responsible for the abuse. This can lead to a lack of necessary support for some victims and the feeling of guilt that arises in other victims. These findings propose that there is a differential cognitive impact of sexual abuse compared to physical abuse in adolescents who have survived interpersonal trauma (Botsford et al. 2018).

In addition, Botsford et al. (2018) inferred from their results that older children have more cognitive distortions related to guilt and that the later the first abuse occurs and the closer they are to the aggressor, the more blocking points in the guilt category are presented by the child. The authors, through their clinical experience, state that adolescents who are victimized at an older age feel guilty for not having defended themselves against the abuser and, on the contrary, adolescents who are abused at an earlier age realize that they could not defend themselves.

This systematic review is not without limitations. Although an exhaustive and systematic search was carried out using strict criteria, there is a possibility that some relevant studies were not included due to their unavailability or inaccessibility. In addition, it is also possible that studies with non-significant results were excluded from this review, given the challenges associated with publishing such results. For these reasons, publication bias is difficult to overcome.

Nevertheless, despite these limitations, procedures were adopted to ensure the methodological rigor required for this type of review, namely the use of Rayyan to organize, manage, and speed up collaboration on the systematic review, as well as the use of EndNote to store, organize, and utilize references, thereby safeguarding the correct referencing of all literature, and the assessment of the quality of the articles, ensuring their methodological robustness.

After a careful analysis of the results of these studies and a respective discussion, this systematic review contributes to developing the knowledge on this form of violence in children and adolescents, particularly in terms of its impact. Thus, sexual abuse has an impact on PTSD in children and adolescents that varies in degree and severity depending on the proximity between the victim and the perpetrator, the way in which the abuse is committed, the characteristics of the abusive experience, the brain changes in victims, and the coping strategies used by the victims. For this reason, the subject is important for clinical practice, as it enables professionals to intervene with the aim of incorporating this experience of victimization into the lives of children and adolescents, ultimately reducing the degree and intensity of PTSD.

Once general results have been presented that prove the impact of this form of victimization on children and adolescents, it would be pertinent for the scientific community to focus their studies on the possible impact of each one of the symptoms of PTSD in this age group. We also consider it important that studies focusing on male participants be carried out, seeking to explore gender differences in victimization. Due to the fact that there is more information on the opposite sex, it is not possible to establish a clear distinction of the level of impact, the way in which the abuse is committed, the characteristics of the abusive experience, the resultant brain changes, or the coping strategies employed by the victims.

Finally, concerning implications for policy, it is crucial to invest in a primary prevention of sexual abuse, particularly among children and adolescents, and educate them to establish bodily boundaries. Alongside this preventive measure, it is essential to invest in the training of all professionals directly involved in addressing this issue, from teachers to magistrates, so that they can identify early signs and behaviors of victimization and prevent secondary victimization.

**Author Contributions:** Conceptualization, A.C.A., M.L. and D.M.; methodology, A.C.A., M.L., A.I.S. and D.M.; validation, A.I.S. and D.M.; formal analysis, A.C.A. and M.L.; investigation, A.C.A., M.L.,

A.I.S. and D.M.; data curation, A.C.A. and M.L.; writing—original draft preparation, A.C.A. and M.L.; writing—review and editing, A.I.S. and D.M.; supervision, A.I.S. and D.M. All authors have read and agreed to the published version of the manuscript.

**Funding:** This work was partially financed by national funds through the Foundation for Science and Technology (FCT) within the framework of the Research Centre for Child Studies (CIEC) of the University of Minho projects under the references UIDB/00317/2020 and UIDP/00317/2020.

**Institutional Review Board Statement:** Not applicable.

**Informed Consent Statement:** Not applicable.

**Data Availability Statement:** The data presented in this study are available upon request from the corresponding author.

**Conflicts of Interest:** The authors declare no conflicts of interest.

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
