# Peer review of "Impact of Sexual Abuse on Post-Traumatic Stress Disorder in Children and Adolescents: A Systematic Review"

_socsci, doi:10.3390/socsci13040189_

Round 1

Reviewer 1 Report

Comments and Suggestions for Authors

After paper review:
Dear researcher(s), let me first thank you for your work and for your interest in such
important topic. Paper was swell and I very much enjoyed reading your work.
Here are my suggestions:

1) title is good

2) You set precise goals and you managed to meet your goals within your research

3) Excellent use of English language. One thing aboute introduction: "According to Cruz et al. 
(2021), in a report that compiles data from several countries in the United Kingdom be
tween 2016 and 2017, points to 54,846 thousand reports of sexual violence against minors 
and perspectives the increase of cases over the years". This sentence might use some fixing.

4) Introduction is engaging and precise, and makes people want to keep reading even though it concerns such a delicate topic.

5) Method (including eligibility criteria, search strategy, study selection, data extraction) is good. Tables are well built. I suggest you try and find more researches within the last 5 years (only five researches were conducted within the last 5 years).

6) Results section is well written, although table 2 and table 3 could have been developed in a different way. Nonetheless, you meet your goals. 

7) References are relevant and goal oriented, you just need to use more updated researches; as for method, most of the references are not from the last 5 years, you should try and find some more actual researches.

Author Response

Reviewer #1:

Question 3): One thing about introduction: "According to Cruz et al. (2021), in a report that compiles data from several countries in the United Kingdom between 2016 and 2017, points to 54,846 thousand reports of sexual violence against minors and perspectives the increase of cases over the years". This sentence might use some fixing.

Answer 3): We appreciate your comment, and the sentence has been corrected (please see the introduction, lines 16, 17, 18 and 19).

Question 7): References are relevant and goal oriented, you just need to use more updated researches; as for method, most of the references are not from the last 5 years, you should try and find some more actual researches.

Answer 7): We are grateful for this suggestion, which seems to us to be pertinent and justified. In order to respond to your suggestion, we're running the search expression again to see if any new articles have appeared. As a result of this search, 3 articles emerged: 1 of them was a duplicate, 1 was inaccessible and 1 did not meet the inclusion criteria. Please check figure 1 and the paragraph that precedes it.

Reviewer 2 Report

Comments and Suggestions for Authors

The presented manuscript analyzes a topic of special interest in the field of psychology. Therefore, I think this is a relevant topic.

In the course of reading the manuscript, a main question arises: Why is the most current publication from 2018? If this work is finally published, the date will be 2014. There are six years of difference.

Regarding certainty in the evidence, present evaluations of the certainty (or confidence) in the body of evidence for each outcome evaluated.

Regarding the discussion, the authors should improve all the sections below:

1. Provide a general interpretation of the results in the context of other evidence.

2. Argue the limitations of the evidence included in the review.

3. Argue the limitations of the review processes used.

4. Argue the implications of the results for practice, policy, and future research.

Author Response

Reviewer #2:

Question: In the course of reading the manuscript, a main question arises: Why is the most current publication from 2018? If this work is finally published, the date will be 2014. There are six years of difference.

Answer: We are grateful for your concern, which seems to us to be pertinent and justified. In order to respond to your suggestion, we're running the search expression again to see if any new articles have appeared. As a result of this search, 3 articles emerged: 1 of them was a duplicate, 1 was inaccessible and 1 did not meet the inclusion criteria. Please check figure 1 and the paragraph that precedes it.

Question 1), 2), 3) and 4):  1. Provide a general interpretation of the results in the context of other evidence; 2. Argue the limitations of the evidence included in the review; 3. Argue the limitations of the review processes used; 4. Argue the implications of the results for practice, policy, and future research.

Answer to all of this questions: We understand your concern, so we improve the limitations of the study elaborating on the points raised (please see the last five paragraphs of the discussion).